# Neuroprotective Activities of Curcumin in Parkinson’s Disease: A Review of the Literature

**DOI:** 10.3390/ijms222011248

**Published:** 2021-10-18

**Authors:** Eslam El Nebrisi

**Affiliations:** Department of Pharmacology, Dubai Medical College, Dubai 20170, United Arab Emirates; dr.eslam@dmcg.edu; Tel.: +971-4212-0556

**Keywords:** curcumin, Parkinson’s disease, neuroprotection, anti-inflammatory, antioxidant, α7-nAChR

## Abstract

Parkinson’s disease (PD) is a slowly progressive multisystem disorder affecting dopaminergic neurons of the substantia nigra pars compacta (SNpc), which is characterized by a decrease of dopamine (DA) in their striatal terminals. Treatment of PD with levodopa or DA receptor agonists replaces the function of depleted DA in the striatum. Prolonged treatment with these agents often has variable therapeutic effects and leads to the development of undesirable dyskinesia. Consequently, a crucial unmet demand in the management of Parkinson’s disease is the discovery of new approaches that could slow down, stop, or reverse the process of neurodegeneration. Novel potential treatments involving natural substances with neuroprotective activities are being developed. Curcumin is a polyphenolic compound isolated from the rhizomes of Curcuma longa (turmeric). It has been demonstrated to have potent anti-inflammatory, antioxidant, free radical scavenging, mitochondrial protecting, and iron-chelating effects, and is considered a promising therapeutic and nutraceutical agent for the treatment of PD. However, molecular and cellular mechanisms that mediate the pharmacological actions of curcumin remain largely unknown. Stimulation of nicotinic receptors and, more precisely, selective α7 nicotinic acetylcholine receptors (α7-nAChR), have been found to play a major modulatory role in the immune system via the “cholinergic anti-inflammatory pathway”. Recently, α7-nAChR has been proposed to be a potential therapeutic approach in PD. In this review, the detailed mechanisms of the neuroprotective activities of curcumin as a potential therapeutic agent to help Parkinson’s patients are being discussed and elaborated on in detail.

## 1. Introduction

Parkinson’s disease (PD) is the second most common neurodegenerative disease after Alzheimer’s disease (AD), which was first described by an English physician and surgeon, James Parkinson, who wrote his Essay on the Shaking Palsy in 1817, and was later named Parkinson’s disease by Jean-Marie Charcot [1]. PD is a slowly progressive multisystem disorder rather than just a disease, involving massive neuropathological degeneration in dopaminergic neurons of the SNpc and their terminals in the striatum.

Pathologically, the disease is distinguished by the phosphorylation of alpha-synuclein protein and the formation of proteinaceous inclusions, Lewy bodies (LB) in neurons and Lewy neurites (LN) in axons and dendrites, as well as dopaminergic nigrostriatal neuronal degeneration [2]. Mechanistically, several factors have been implicated in dopaminergic neuronal degeneration: (1) A genetic mutation causes protein misfolding and oxidative stress. (2) Exposure to toxins results in mitochondrial dysfunction and an increase in reactive oxygen species (ROS). (3) Neuroinflammation and chronic microglial activation, both of which cause neuronal degeneration by releasing pro-inflammatory mediators and altering other molecular and cellular functions [3,4,5].

PD is an age-related disorder, where the prevalence of the disease increases with advancing age. In industrialized countries, the prevalence is around 1% for people over the age of 60 and 0.3 percent for people of all ages [6]. Although the vast majority of cases are sporadic, about 10–15 percent of patients have a positive family history of PD. Environmental insults, among other factors, contribute to the degenerative changes seen in PD, including mitochondrial dysfunction, oxidative stress, changes in protein handling, immune-modulator adaptations, and alterations to other molecular and cellular functions [3,7].

To date, no drug cures or stops the progression of PD. Being mainly dysfunctional in the dopaminergic system in the brain, Levodopa or L-dopa (L-3,4-dihydroxyphenylalanine) were introduced in the 1960s as a prodrug of dopamine (DA) which enhances intracerebral DA concentration. Since its approval by the FDA in 1970, L-dopa has been the gold standard treatment for PD. However, after several months to years of treatment with L-dopa, patients develop adverse effects such as dyskinesias [8,9], which are known as L-dopa-induced dyskinesias (LIDs). With the limitation of L-dopa use, other strategies have been implemented to enhance dopamine release, such as DA agonist, monoamine oxidase type B inhibitors (MAO-B), catechol-*O*-methyl transferase inhibitors (COMTIs), anticholinergic, beta-blocker, antipsychotic, and amantadine [10]. Surgical intervention becomes an option with deep brain stimulation (DBS) as a direct effect in selected PD patients [11,12]. Typically, all the available drugs are designed to replace the function of depleted DA in the striatum without any neuroprotective activity. However, prolonged treatment with these agents often has variable therapeutic effects and leads to the development of undesirable adverse reactions. With time, treatment efficacy starts to decline and patients’ symptoms and disability get worse, affecting the quality of life with the need for home care and frequent hospital admission [13,14]. Based on several studies, the life expectancy of PD patients, after the onset of the disease, ranges from 6.9 to 14.3 years [15].

As previously stated, mitochondrial dysfunction, oxidative stress, and modifications in protein handling are the three main pathophysiological derangements in PD which affect cellular functions [2,3,5]. Thus, to ensure fewer side effects and to target different intracellular signaling pathways, a multidisciplinary approach is needed that employs several drugs or compounds at minimally effective doses. Natural polyphenol compounds derived from plants, such as curcumin, have many favorable biological properties. Curcumin is emerging as a promising candidate for the use of innovative strategies of natural molecules with neuroprotective properties as adjuvant therapy in Parkinson’s disease.

In this context, this review focuses on the neuroprotective activities of curcumin in PD and the various mechanisms involved. Curcumin’s pharmacokinetics, pharmacodynamics, biological, cellular, and molecular properties are all addressed. A special emphasis is given to curcumin’s neuroprotective activities via a α7-nAChR-mediated mechanism, safety profile, current and upcoming clinical trials for clinical application.

## 2. Curcumin as a Potential Neuroprotective Agent

Curcumin was named after Vogel and Pelletier, the first to isolate a “yellow coloring-material” from the rhizomes of Curcuma longa in (turmeric) 1815. Later, in 1842, they discovered that turmeric is a complex mixture of ingredients and were successful in isolating pure curcumin oil. In 1910, Milobedeska and Lampe characterized its structure as diferuloylmethane, or 1,6-heptadiene-3,5-dione-1,7-bis (4-hydroxy-3-methoxyphenyl) (Figure 1), and three years later they synthesized curcumin [16].

### 2.1. Chemical and Physical Properties of Curcumin

Curcumin is a symmetric molecule composed of three major chemical entities: two aromatic ring systems containing *O*-methoxy phenolic groups linked by a seven-carbon linker containing α, β-unsaturated diketone moiety (Figure 2). Curcuminoid (the yellow-pigmented turmeric preparation) accounts for 3–5 percent of turmeric and is primarily composed of three derivatives: curcumin (diferuloylmethane, curcumin I ~77%), demethoxycurcumin (DMC, curcumin II), bisdemethoxycurcumin (BDMC, curcumin III), and cyclo-curcumin [17,18]. All three derivatives are considered to be natural turmeric analogs. Curcumin exhibits keto-enol tautomerism, with enol forms predominating in alkaline media and keto forms predominating in acidic or neutral media [17]. Curcumin is a hydrophobic compound that is insoluble in polar or neutral solvents such as water. It can be dissolved in organic or hydrophobic solvents such as dimethylsulfoxide (DMSO), ethanol, and acetone [19]. Tetrahydrocurcumin (THC), dimethyl curcumin, di-demethyl curcumin, Vanillylidenacetone, Di-(tert-butyl-dimethylsilyl) curcumin, *O*-tert-butyl-dimethylsilyl curcumin, and curcumin-d6 are all commercially available curcumin metabolites.

### 2.2. Pharmacokinetics and Pharmacodynamics of Curcumin

Human studies of curcumin’s pharmacokinetics yielded results that were similar to those obtained from animal studies. Because of its poor absorption, curcumin has a low bioavailability in plasma and tissues, rapid hepatic metabolism, as well as rapid systemic elimination through the gut with a peak human plasma level of 0.41–1.75 µmol/L after the oral administration of 4–8 g of curcumin [20,21]. Many studies have shown that curcumin is primarily metabolized in the liver, where it undergoes extensive reduction via alcohol dehydrogenase, followed by glucuronate and sulfate conjugation [8,21]. Furthermore, Perkins and colleagues reported that humans require a daily dose of 1.6 g curcumin to achieve the desired results [22].

Almost all studies have confirmed that unformulated curcumin has low bioavailability in both animals and humans [23,24]. Various formulations have been developed to improve curcumin bioavailability. Nano curcumin, for example, was developed to improve curcumin solubility in an aqueous solution. Cheng et al. generated a nanoparticle form of curcumin that resulted in a higher plasma concentration and a six-fold higher AUC with a longer mean residence time in mice brains. [25]. Polylactic-co-glycolic acid (PLGA) and liposomal-formulated curcumins improved water solubility of the compound [26,27,28]. In regards to curcumin permeability, cyclodextrin (CD) encapsulated curcumin improved curcumin permeability compared to unformulated curcumin [29]. Concomitant administration of piperine with curcumin significantly reduced elimination and half-life clearance of curcumin [23,24]. Alginate–curcumin nanoparticles (Alg-NP-Cur) [30], glyceryl mono-oleate nanoparticles loaded with piperine and curcumin (GMO-NP-Pip/Cur) [31], curcumin-loaded lactoferrin nanoparticles (Lf-NP-Cur) [32], and curcumin-loaded polysorbate 80-modified cerasome (CPC) nanoparticles (NPs) [33], are different preparations developed to maximize curcumin bioavailability.

### 2.3. Biological Properties of Curcumin

Curcumin, a multi-targeted compound, has traditionally been used as a dietary spice and a medicinal herb in Asian countries for a variety of pathologies due to its anti-inflammatory properties [34], and antioxidant properties [35,36]. Moreover, curcumin possesses antibacterial [37], antiviral [38], antifungal [39], anti-arthritic [40], hepatoprotective [41], anti-thrombotic [42], cardio-protective [43], hypoglycemic [44], anti-allergic [45,46], wound-healing [47], and chemo-preventive and anticancer properties [48,49,50]. Curcumin’s anti-inflammatory and antioxidant effects, among others, form the basis of curcumin’s critical neuroprotective effects in a variety of neurological diseases affecting both the central and peripheral nervous systems. Several molecular targets of curcumin have been identified based on extensive evidence from in vitro and in vivo studies.

### 2.4. Molecular and Cellular Neuroprotective Mechanisms of Curcumin in PD

The present review focuses on recent advances and the mechanisms underlying the wide range of biological effects of curcumin against neurodegenerative diseases, specifically Parkinson’s disease. Curcumin’s ability to modulate the functions of multiple signal transduction pathways has been linked to a reduction in disease progression. Curcumin interacts with transcription factors such as z transcription (STAT) proteins [51], growth factors and their receptors, e.g, epidermal growth factor receptors and HER2 [52,53], cytokines, e.g., interleukin 1b (IL-1b), interleukin 6 (IL-6) [54], enzymes, e.g., hemox (HO-1) [55], and genes that regulate cell proliferation and apoptosis [56]. The ability of curcumin to modulate and interact with multiple cell signaling pathways and proteins strongly indicates that this polyphenol is an effective multi-targeted compound [57,58,59]. This conclusion is in line with several recently published reports identifying curcumin as a potent epigenetic regulator [60,61]. Interestingly, curcumin’s inhibitory effect on MOA-B enzyme [62], which would lead to an increase in the level and availability of DA in the brain, has gained much attention in recent years, as discussed below.

A critical unmet need in the management of PD is the discovery of new approaches that could slow, stop, or ideally reverse, the process of neurodegeneration. Curcumin’s neuroprotective potential has been demonstrated in several recent studies using various animal models of Parkinson’s disease [63,64,65,66,67,68,69,70]. For instance, Zbarsky described the protective effects of curcumin on the number of TH-positive neurons as well as on striatal DA level and its metabolites; dihydroxyphenylacetic acid (DOPAC) and homovanilic acid (HVA) against 6-hydroxydopmine (6-OHDA) induced neurodegeneration in animal models of PD [71]. The advantage of curcumin over other derivatives, such as demethoxycurcumin (DMC) and bisdemethoxycurcumin (BDMC), was reported on DA receptor (D2) binding activities and on the number of TH +ve neurons [72]. Yang et al. described the protective effects of curcumin on the injured hippocampus in an 6-OHDA model of PD, including a significant improvement in mental status, weight gain, neurobehaviors, learning and memory, levels of dopamine and norepinephrine, neural regeneration in hippocampal tissue, and cell survival-related signaling pathways such as BDNF, TrkB, and PI3K [73]. Moreover, brain-derived neurotrophic factor (BDNF), a member of the neurotrophin growth factor family, which is involved in various neurological functions, is affected in PD [74]. Curcumin restores neuronal regeneration by stimulating Trk/PI3K signaling cellular cascade, reducing levels of tumor necrosis factor-α (TNF-α) and caspase activity, hence increasing levels of BDNF in 6-OHDA model of PD [73,75]. Recently, we investigated the neuroprotective effects of curcumin in a 6-OHDA animal model of PD [70]. The results indicated that curcumin enhances the survival of striatal TH fibers and SNpc neurons, decreases abnormal turning behavior, and exerts neuroprotective properties at least partly via an α7- nAChR-mediated mechanism. These findings provide evidence that α7-nAChRs could be a potential therapeutic target and curcumin would be the first natural agent which is reported to modulate nicotinic receptors in PD.

#### 2.4.1. Curcumin Anti-Inflammatory Effects

Inflammation is an adaptive physiological process by which our bodies fight against injuries or infections, and trigger a host–immune response. Inflammation plays a major role in several pathological conditions including neurodegenerative (PD and AD), autoimmune, cardiovascular, endocrine, and neoplastic disorders [76,77]. It is a complex interaction that aims at removing the invading agent or damaged tissue by the activation of various inflammatory mediators. Over-activation of the immune system and inflammatory responses may cause further tissue damage [78,79]. Neuroinflammation has been linked to neurodegenerative diseases, including PD, but whether neuroinflammation is a trigger or a result of neuronal loss remains controversial [78,79]. Current advances in molecular biology provide evidence that neuroinflammation plays an important role in the pathogenesis of PD [80,81]. Immune reactions in the form of glial activation and inflammatory processes may also participate in the cascade of events, leading to neuronal degeneration in PD. Activated microglia expresses various cell-surface receptors, leading to increased levels of cytokines such as TNF-α, interleukin-1β (IL-1β), and interferon-γ in the substantia nigra of PD patients [82]. These enforce chronic inflammation of the brain, neuronal dysfunction, and neurodegenerative loss in PD [79,82]. Remarkably, curcumin exhibits anti-inflammatory activities by inhibiting inflammatory cytokines, interleukins (ILs), chemokines, as well as inflammatory enzymes, cycloxygenase-2 (COX-2), GFAP level, and cyclin D1 [83,84]. Additionally, curcumin suppresses the expression of inducible nitric oxide protein (iNOS mRNA expression), LPS-induced TNF-α, IL-1β, IL-6 production, and JNK phosphorylation, collectively inhibiting cell apoptotic pathway and enhancing survival [85,86]. Interaction with and modulation of the effects of various inflammatory mediators by curcumin verifies its anti-inflammatory properties [16,65].

#### 2.4.2. Curcumin Antioxidant Effects

Oxidative stress plays a major role in acute, chronic, and degenerative diseases. Oxidative stress results from an imbalance between the formation and neutralization of reactive oxygen species in our bodies, leading to the generation of free radicals and energy failure [87]. The progressive dopaminergic neurotoxicity in SNpc has been directly linked to oxidative stress as a major element in the degenerating cascade underlying neuronal degeneration in PD. ROS oxidative stress is explicitly related to mitochondrial enzyme dysfunction of the respiratory chain, namely, complex I, which results in the majority of detrimental neuronal degeneration in PD [5,65]. Additionally, the abundance of polyunsaturated fatty acids in the brain, which undergoes lipid peroxidation in oxidative stress, liberates more toxic by-products. Besides, the damaging effects of reactive nitrogen species such as nitric oxide (NO) and peroxynitrite on several steps of dopamine synthesis, mitochondrial dysfunction, and consequently dopaminergic cell aging and death in PD, have been reported [88,89]. The potent activity of curcumin against pro-oxidants such as superoxide radicals, hydrogen peroxide, and nitric oxide radicals, as well as enhancing anti-oxidant enzymes such as catalase, superoxide dismutase (SOD), glutathione peroxidase (GPx), and heme oxygenase-1 (OH-1), result in a decrease in lipid peroxidation and organ damage [90,91,92]. Through its antioxidant effects, Song et al. reported that curcumin has restorative effects on degenerated neurons in substantia nigra, and produces marked improvement in the motor, cellular, and biochemical alterations in PD rats [93]. Likewise, Khawaja provided extensive evidence of the potent activity of curcumin against pro-oxidants such as superoxide radicals and hydrogen peroxide radicals, as well as enhancing antioxidant enzymes such as catalase, superoxide dismutase (SOD), and glutathione peroxidase (GPx), which result in a decrease in lipid peroxidation and subsequently neuronal damage in SNpc in a 6-OHDA model of PD [94]. Similar findings of the antioxidant neuroprotective properties of curcumin and, to a lesser extent, other curcuminoid derivatives such as demethoxycurcumin and bisdemethoxycurcumin, were later confirmed [72]. Furthermore, curcumin antioxidant activities restored dopamine as well as tyrosine hydroxylase levels in an MPTP model of PD [95]. One of the main elements in the development of the nervous system and regulation of brain neurogenesis is the activation of the Wnt/β-catenin signaling pathway [96]. Curcumin has been shown to protect against oxidative stress-induced neurodegeneration in 6-OHDA PD by stimulating the Wnt/β-catenin pathway, which consequently leads to improving cell viability, survival, and reducing neuronal apoptosis [97]. Modification of the downstream cellular mediators, such as c-Myc and cyclin D1 in the Wnt signaling cascade, could also play a significant role in the neuroprotective activities of curcumin [98]. The methoxy and phenolic groups on the benzene rings and the β-diketone moiety in the curcumin structure (Figure 2) are thought to be essential for its anti-oxidant properties [17,99]. Interestingly, curcumin exhibits a stronger anti-oxidant activity even when compared with vitamin C and E [92].

#### 2.4.3. Curcumin Free Radicals’ Scavenging Activities

Oxygen is the motive force for most of the irreversible cellular injury and neurodegenerative changes occurring in PD. Although oxygen is very fundamental for any living system, it is inherently harmful at the same time, a phenomenon known as “the oxygen paradox” [100,101]. Preliminary evidence for the role of the oxygen paradox in PD was strongly supported by post-mortem brain analysis of PD patients demonstrating high levels of oxidized DNA, protein, and lipid [102,103]. The theory behind the oxygen paradox relies on the deterioration of cellular scavenging activity, with eventual protein carbonylation, formation of nitrotyrosine, and subsequent protein aggregation [104,105]. In support of that, pathological protein aggregates such as α-synuclein, the ubiquitin-proteasome system (UPS), and chaperones have been reported in PD [106]. Curcumin comprises several functional groups responsible for its antioxidant activity. In addition, curcumin can directly scavenge reactive molecules and break the oxidation chain [107]. Curcumin treatment significantly reduces carbonylated protein and nitrotyrosine-modified proteins in the rotenone-induced model of PD [104]. ROS consists of both free radical oxidants and non-radical molecular oxidants. Free radical oxidants participate in single electron transfer reactions and hydrogen atom abstraction. The three active sites, methoxy and phenolic groups on benzene rings and the β-diketone moiety of curcumin, can undergo oxidation by electron transfer and hydrogen abstraction, and thus form stabilized phenoxyl radicals. Curcumin is an excellent scavenger for most ROS in a concentration or dose-dependent manner [92]. Remarkably, curcumin inhibits oligomerization of α-synuclein, protein aggregation, and consequently neural toxicity [65,108], and produces potential inhibitory effects on astrocytic activation as well as NADPH oxidase system [65]. Regenerating the oxidative status of curcumin could be achieved by a chain-breaking or a hydrogen donor antioxidant, such as vitamin E or ascorbic acid (Figure 3).

#### 2.4.4. Mitochondrial Protection

Mitochondria play a central role in maintaining cellular homeostasis [100]. Neuronal cells are highly dependent on mitochondrial energy production [109,110]. Extensive data from cellular, genetic, toxin-induced animal studies, and postmortem human brain demonstrate mitochondrial dysfunction in PD in the form of inhibited complex I and subsequent mitochondrial electron chain inhibition, energy failure, oxidative stress, and dopaminergic cell death in PD [3,109,111,112,113]. Curcumin as a multi-targeted compound that can serve as a neuroprotective agent. Oral administration of curcumin protects Swiss albino mice against rotenone-induced dysfunction in the mitochondrial respiratory chain and conserves the mitochondrial enzyme complex, which is reflected in the improvement of motor behavior of the animals after three weeks of curcumin administration [114]. In addition, curcumin beneficially modulates mitochondrial malfunction and immature senescence [115,116]. It efficiently improves mitochondrial enzyme complex activities in the rotenone-induced PD model [114]. Furthermore, in the PTEN-induced putative kinase 1 (*PINK1*), a genetically mutated model of PD in mice, pre-treatment with curcumin improves cell viability, improves mitochondrial membrane potential, and reduces apoptosis in SH-SY5Y neuroblastoma cells [117].

#### 2.4.5. Curcumin Iron-Chelating Properties

Iron is required for several fundamental functions in the brain. Iron homeostasis management entails controlling iron influx, efflux, and storage. Metals such as iron (Fe), zinc (Zn), and copper (Cu) accumulate in the brain as we age [118]. With increasing age, there is an increase in brain iron concentration and deposition as a result of iron mismanagement, resulting in oxidative injury and neuronal degeneration [119]. Iron is either stored in lysosomes or bound to neuromelanin and ferritin in neuronal cells. The latter is a bio-vital chelator that is regulated by the mitochondria [120,121]. Iron deposition in neuronal cells has also been considered as one of the major findings in postmortem PD brains, including substantia nigra [122,123]. Notably, the iron-chelating activity of curcumin has been previously described [124]. Du et al. successfully demonstrated a decrease in iron-positive cells following curcumin treatment in 6-OHDA induced model of PD [125]. Supporting evidence of combined curcumin treatment and desferrioxamine, a potent iron-chelating agent, reflects the protective effect of curcumin on dopaminergic neuronal loss in the PD model [126]. The use of deferoxamine in conjunction with a novel delivery system; curcumin-loaded-nanocarrier in the rotenone-induced Parkinson’s disease model was recently supported, such combination provided clear protection for dopaminergic neurons against iron deposition [127]. Sharma and colleagues obtained similar results when inhibiting iron deposition in dopaminergic neurons [65].

## 3. Neuroprotective Mechanisms of Curcumin via Nicotinic Acetylcholine Receptors

Curcumin’s pharmacological actions are thought to be mediated by a variety of ligand-gated ion channels and receptors [128]. The recent study on the effects of the natural polyphenol compound provides evidence that curcumin possesses a potent neuroprotective effect as it preserves the integrity of the nigrostriatal dopaminergic system. This is distinctly manifested in the improved motor behavioral performance in the curcumin-treated animals through a α7-nAChRs-mediated mechanism [70]. This study adds to previous in vitro studies that show that curcumin enhances the effects of acetylcholine (ACh) through the function of α7-nAChRs in a concentration-dependent manner [129]. In addition, the results from another in vitro study highlight the significant role of curcumin in modulating the fluxes of calcium (Ca^2+^) ions via α7-nAChRs [130]. Based on the previous findings that curcumin acts as a type II PAM of α7-nAChRs and a potentiator of receptor function by significantly decreasing desensitization [129], it is reasonable to conclude that curcumin’s PAM action on α7-nAChRs has a beneficial effect in mediating neuroprotective effects [131,132]. Curcumin’s time-tested safety, neuroprotective efficacy, and preliminary clinical success of agents targeting nicotinic receptors in PD make it an appealing natural candidate for further investigation and development in the search for PD therapeutics.

Our in vitro, in silico, and in vivo findings suggest that increasing Ca^2+^ influx may have a neuroprotective mechanism in neuronal and non-neuronal cells via various intracellular mechanisms, as shown in Figure 4 [70,129,130]. Stimulation of presynaptic α7-nAChR stimulates vesicular DA release via a Ca^2+^-dependent facilitation mechanism [133,134,135]. Extracellular signal-regulated mitogen-activated protein kinase (ERK/MAPK) activation can be triggered by protein kinase A (PKA) and/or calcium-calmodulin-dependent protein kinase (CaMK) [136]. A rise in intracellular Ca^2+^ levels is considered as a trigger factor of both signaling cascades. Activation of (ERK/MAPK) is a crucial signaling event in the cell survival pathway via upregulation of the cellular transcription factor; cAMP response element-binding (CREB), increasing gene expression of tyrosine hydroxylase and enhancing DA release [137,138]. α7-nAChR is also expressed on microglia and astrocytes and plays a major role in immune response via the “cholinergic anti-inflammatory pathway”. Activation of α7-nAChR results in an increase in intracellular Ca^2+^ concentration, and consequently modulates Janus kinase 2 (JAK2) and/or signal transducer and activator of transcription 3 (STAT3), ending up with an upregulation of protein kinase B (PKB), leading to inhibition of nuclear factor-kB (NFκB) [139]. The lipid signaling cascade that is started by protein kinase C (PKC), via phosphorylation of phosphatidylinositol 3-kinase (PI3K/Akt), is accredited with modulating the activities of neuroprotective and apoptotic factors, such as Bcl-2 and caspases, respectively [140,141,142]. Recent data demonstrate that the regulation of neuroinflammatory reactions by curcumin occurs through the modulation of the microglial JAK/STAT signaling pathway [143]. Collectively, all or some of these factors result in decreased apoptosis, enhance neuronal survival, modify immune responsiveness, and produce alteration in synaptic plasticity [144].

## 4. An Update and Current Perspectives on Curcumin

α7-nAChR is thought to be a potential nutraceutical agent for a variety of neurological disorders, including Parkinson’s, Alzheimer’s, and schizophrenia. Clinical trials are currently underway for a number of α7-nAChR agonists and modulators [145]. Interestingly, α7-nAChR-positive allosteric modulators (PAMs) demonstrated very positive and promising results. Curcumin, a type II PAM [129], is a natural compound with a high safety profile and has no reported toxicity from in vitro to in vivo, and clinical trials [20,146,147,148,149,150] if administered at the recommended dose [22,151]. Curcumin has undergone several clinical trials for the treatment of neurodegenerative disorder and demonstrated a pro-cognitive effect in rodents and non-human primates [152,153,154,155,156,157,158,159,160,161].

Overall, the current findings of the clinical trial on nicotinic receptors and PD or curcumin and neurodegenerative disorders such as PD are very promising, but more pre-clinical studies and clinical trials are needed to improve curcumin’s bioavailability and define its hidden targets.

## 5. Concluding Remarks

Curcumin is a neuroprotective agent with antioxidant [35,36], anti-inflammatory [86], free radical scavenging [107], mitochondrial protecting [62], and iron-chelating properties [125], which enhance DA levels in the brain [62]. The interaction of curcumin with α7-nACh receptors provides further evidence for a potential neuroprotective role for curcumin in PD. Additionally, curcumin and derivatives show a high safety profile with minimal reported toxicity as demonstrated both in in vitro and in in vivo studies in PD models. Therefore, gaining a better understanding of the neuroprotective properties of curcumin could have significant therapeutic implications. The evidence reviewed supports curcumin’s powerful molecular and cellular effects in neurodegenerative disorders as an appealing strategy for improving PD management and prognosis.

## Figures and Tables

**Figure 1 ijms-22-11248-f001:**
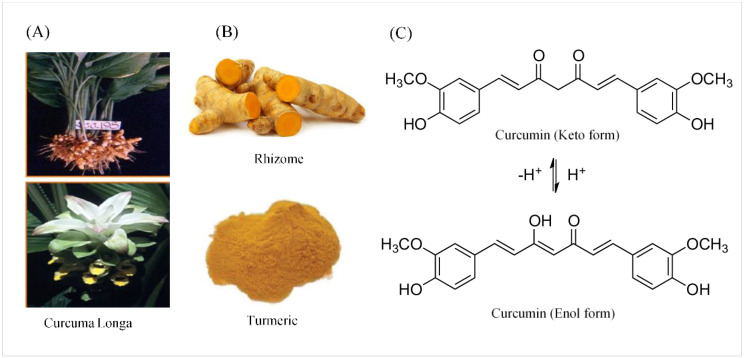
The source, crude form, and chemical structure of curcumin. (**A**) The botanic source of turmeric. (**B**) Crystallized powder of curcumin. (**C**) The enol and keto forms of curcumin.

**Figure 2 ijms-22-11248-f002:**
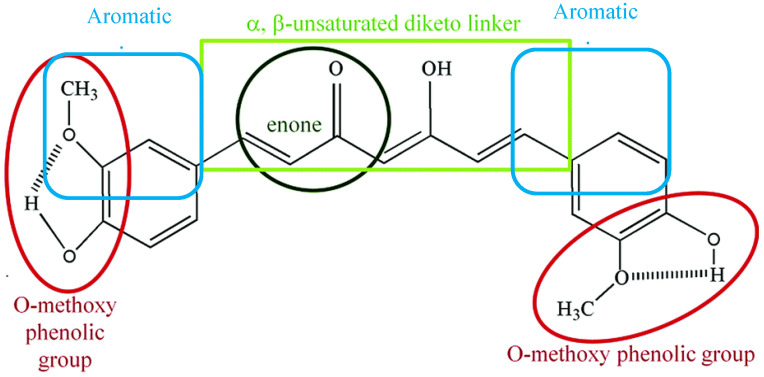
Chemical structural groupings that are responsible for the antioxidant properties of curcumin. Curcumin is composed of three chemical entities: two aromatic ring systems containing *O*-methoxy phenolic groups linked by a seven-carbon linker containing of α, β-unsaturated diketone moiety.

**Figure 3 ijms-22-11248-f003:**
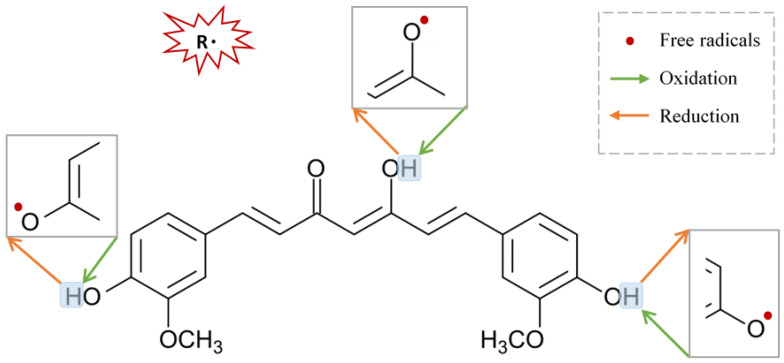
Suggested sites of exchange of phenol OH-group in curcumin structure with free radical oxidants, and its regeneration by a hydrogen donor antioxidant.

**Figure 4 ijms-22-11248-f004:**
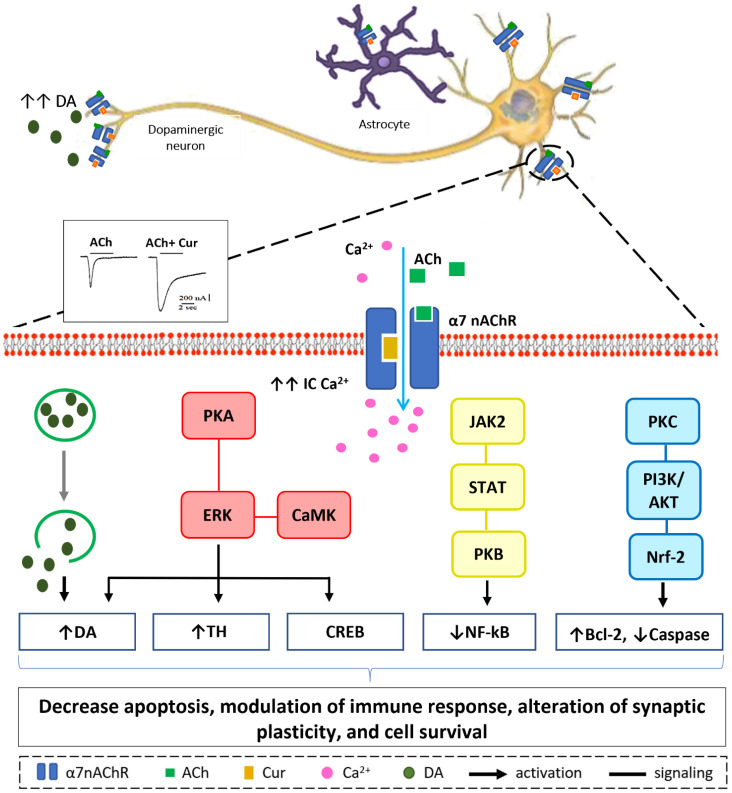
Hypothetical model of Ca^2+^-dependent cell survival mechanism. Curcumin modulate α7-nAChR allosterically allowing more Ca^2+^ entry into the cell as depicted from the electrophysiological recording. Increase in intracellular Ca^2+^ concentration will lead to a cascade of events in dopaminergic neurons (from left to right): Facilitation of dopamine release from synaptic vesicles. Activation of ERK by PKA and/or CaMK, upregulate CREB protein, increase tyrosine hydroxylase activity, and activate dopamine release. JAK2/STAT3 signaling pathway leads to inhibition of NF-kB translocation via PKB activation. Increase in IC Ca^2+^ attenuates inflammatory response in immune cells activating protein kinase C, PKC appears to activate downstream signaling PI3K/AKT pathways that promotes Nrf-2 translocation resulting in modulation of cell survival proteins; Bcl-2 and caspase.

## Data Availability

Not applicable.

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
