# Peer review of "Neuroprotective Activities of Curcumin in Parkinson’s Disease: A Review of the Literature"

_ijms, 2021, doi:10.3390/ijms222011248_

Round 1
Reviewer 1 Report
In my opinion, this topic has comprehensively been covered in various recently published review articles, and this manuscript does not bring any novelty. Another point is that the whole paper needs major reorganization and improvement. The author should update the manuscript by using newly published studies (preclinical and clinical investigations). Here, I have several comments and concerns that might help improve the quality of the manuscript.
- I recommend the author replace the section on Parkinson’s disease and its subsections with a section named ''Introduction'', where information about the disease, current available pharmacologic therapies, and their limitations, the need for finding new sources of drugs such as natural products along with the aim of this review should be presented.
- Section (2.3. Biological Properties of Curcumin). This section needs more refinements, where some biological properties are missed. For example, curcumin exposed antiviral activities against various types of herpesvirus infections and their associated inflammation. I recommend the author use a newly published report (Nutraceutical Curcumin with Promising Protection against Herpesvirus Infections and Their Associated Inflammation: Mechanisms and Pathways. Microorganisms. 2021 Jan 31;9(2):292. doi: 10.3390/microorganisms9020292), where this information is mentioned.
- Molecular and cellular mechanisms of curcumin should be gathered in one section. Both mechanisms are biologically connected and it is better to mention them in one section. Also, I recommend the author make a table or figure to present these mechanisms. This will help the readers to go directly into the point and extract the desired information.
- Figures 2,3, and 4. All reported information in these figures should be clarified by providing scientific proof (for instance, references that have reported such data).
- I recommend the author add a new section that provides strategies/technologies that might improve the activity of curcumin as a promising drug for Parkinson’s disease.
- Again, I recommend the author use newly published reports to provide up to date review article as well as novelty.
- Finally, I recommend the author double-check the whole manuscript for grammatical and typing errors.
Author Response
Dear Ms. Corrine Gao
Manuscript ID: ijms-1361603
Thank you for your email regarding my manuscript entitled: “Neuroprotective activities of curcumin in Parkinson’s disease: A review of the literature “. I appreciate the time you took to thoroughly review the paper and write the comments and think that it deserves revision and resubmission.
I read through all of the comments and attempted to respond to each one individually. However, I think that the answer to comment No. 5 from reviewer 1 can be refreshed and expanded upon further and the whole paper once accepted.
I would be pleased to hear from you and make any additional changes required to strengthen the paper and/or enable publication in your prestigious journal.
Thanks again for your time and effort. I look forward to hearing from you.
Sincerely,
Eslam El Nebrisi
Assistant professor - Pharmacology Dept.
Dubai Medical College
P.O Box: 20170, Dubai - United Arab Emirates
Tel: 04 2120555 Ext: 556
Response to Reviewer 1 Comments
Comments and Suggestions for Authors
In my opinion, this topic has comprehensively been covered in various recently published review articles, and this manuscript does not bring any novelty. Another point is that the whole paper needs major reorganization and improvement. The author should update the manuscript by using newly published studies (preclinical and clinical investigations). Here, I have several comments and concerns that might help improve the quality of the manuscript.
Point 1: I recommend the author replace the section on Parkinson’s disease and its subsections with a section named ''Introduction'', where information about the disease, current available pharmacologic therapies, and their limitations, the need for finding new sources of drugs such as natural products along with the aim of this review should be presented.
I'd like to thank the reviewer for this insightful comment. I have now made the suggested changes to the first section and used the suggested sections. In addition, a new additional paragraph has been developed to clarify the type of the manuscript and specify its aims. This paragraph is presented on Page, Para, Lines [Lines 123-127: In this context, this review focuses on the neuroprotective activities of curcumin in Parkinson's disease and the various mechanisms involved. Curcumin's pharmacokinetics, pharmacodynamics, biological, cellular, and molecular properties are all addressed. A special emphasis is given to curcumin’s neuroprotective activities via α7-nAChR mediated mechanism, safety profile, current and upcoming clinical trials for clinical application].
Point 2: Section (2.3. Biological Properties of Curcumin). This section needs more refinements, where some biological properties are missed. For example, curcumin exposed antiviral activities against various types of herpesvirus infections and their associated inflammation. I recommend the author use a newly published report (Nutraceutical Curcumin with Promising Protection against Herpesvirus Infections and Their Associated Inflammation: Mechanisms and Pathways. Microorganisms. 2021 Jan 31;9(2):292. doi: 10.3390/microorganisms9020292), where this information is mentioned.
I appreciate the reviewer's suggestions; I have now refined the section on biological properties, though they are endless. The section on curcumin's biological properties has been updated. In addition, recent references have been used and cited, including the one suggested by the Reviewer [line 188].
Point 3: Molecular and cellular mechanisms of curcumin should be gathered in one section. Both mechanisms are biologically connected and it is better to mention them in one section. Also, I recommend the author make a table or figure to present these mechanisms. This will help the readers to go directly into the point and extract the desired information.
The reviewer's point is well taken, and I value his perspective. Both sections have been merged. Moreover, I have added a table of contents and a table of figures [page 2].
Point 4: Figures 2,3, and 4. All reported information in these figures should be clarified by providing scientific proof (for instance, references that have reported such data).
I would like to emphasize to the Reviewer that the reported information on these figures has been produced by the Author in my Ph.D. figures (1, 2, & 4), which was successfully accepted by UAE University, CMHS-November, 2018). Figure 3 was produced recently during the writing of the manuscript.
Point 5: I recommend the author add a new section that provides strategies/technologies that might improve the activity of curcumin as a promising drug for Parkinson’s disease.
Thanks to reviewer for his valid input. I have formulated and enclosed a full paragraph which is presented on page 11, lines 405-419 under the title: [4. An update and current perspectives on curcumin: α7-nAChR is thought to be a potential nutraceutical agent for a variety of neurological disorders, including Parkinson's, Alzheimer's, and schizophrenia. Clinical trials are currently underway for a number of α7-nAChR agonists and modulators [147]. Interestingly, α7-nAChR positive allosteric modulators (PAMs) demonstrated very positive and promising results. Curcumin, a type II PAM [131], is a natural compound with a high safety profile and no reported toxicity from in vitro to in vivo, and clinical trials [148]–[153] if administered at the recommended dose [22], [154]. Curcumin has undergone several clinical trials for the treatment of neurodegenerative disorder and demonstrated a pro-cognitive effect in rodents and non-human primates [155]–[160], [160]–[162], [163, p.], [164].
Overall, the current findings of the clinical trial on nicotinic receptors and PD or curcumin and neurodegenerative disorders such as PD are very promising, but more pre-clinical studies and clinical trials are needed to improve curcumin's bioavailability and define its hidden targets.].
Point 6: Again, I recommend the author use newly published reports to provide up to date review article as well as novelty.
Kindly note that I have replaced/updated most -if not all- of the old references, with around two-thirds of the total number of references (107) from the last five years (2016-2021).
Point 7: Finally, I recommend the author double-check the whole manuscript for grammatical and typing errors.
In response to the Reviewer comment a thorough grammar, English Language phrasing and typing checks have been undertaken to produce the current version of the paper.
Reviewer 2 Report
Following the analysis of the manuscript titled "Neuroprotective activities of curcumin in Parkinson’s disease: A review of the literature", I appreciate the article's topic is interesting, the presentation of the information is clear and properly structured, and the figures are expressive and reinforce the presented notions. I recommend that it should be revised taking into account the following observations:
- Please update the bibliography, because many of the articles in the list have been published for more than 10 years.
- In the main text clarify the type of manuscript and what was the purpose of this article.
- Please do a thorough check of the abbreviations, some are repeated or missing (e.g. line 31 then line 34; lines 49, 163, 166, 311, 366).
- As the single author of the manuscript please rewrite statements like "In this review, we are discussing and elaborating", "Our recent study on the effects of the natural polyphenol compound provides evidence", "Our in vitro, in silico, and in vivo findings suggest that".
Author Response
Dear Ms. Corrine Gao
Manuscript ID: ijms-1361603
Thank you for your email regarding my manuscript entitled: “Neuroprotective activities of curcumin in Parkinson’s disease: A review of the literature “. I am happy to hear that the reviewers are interested in the topic and think that it deserves revision and resubmission.
I appreciate the time you took to thoroughly review the paper and write the comments. I read through all of the comments and attempted to respond to each one individually 'see attached'. I hope that these changes fully address the concerns of the reviewers and that the manuscript is now in an acceptable form for publication in the IJMS.
I would be pleased to hear from you and make any additional changes required to strengthen the paper and/or enable publication in your prestigious journal.
Thanks again for your time and effort. I look forward to hearing from you.
Sincerely,
Eslam El Nebrisi
Assistant professor - Pharmacology Dept.
Dubai Medical College
P.O Box: 20170, Dubai - United Arab Emirates
Tel: 04 2120555 Ext: 556
Response to Reviewer 2 Comments
Comments and Suggestions for Authors
Following the analysis of the manuscript titled "Neuroprotective activities of curcumin in Parkinson’s disease: A review of the literature", I appreciate the article's topic is interesting, the presentation of the information is clear and properly structured, and the figures are expressive and reinforce the presented notions. I recommend that it should be revised taking into account the following observations:
Point 1: Please update the bibliography, because many of the articles in the list have been published for more than 10 years.
I would like to take this opportunity to thank the reviewer for evaluating the paper as “an interesting and clear work”. Kindly note that I have replaced/updated most -if not all- of the old references, with around two-thirds of the total number of references (107) from the last five years (2016-2021).
Point 2: In the main text clarify the type of manuscript and what was the purpose of this article.
I'd like to thank the reviewer for this insightful comment. A new additional paragraph has been developed to clarify the type of the manuscript and specify its aims. This paragraph is presented in [Lines 123-127: In this context, this review focuses on the neuroprotective activities of curcumin in Parkinson's disease and the various mechanisms involved. Curcumin's pharmacokinetics, pharmacodynamics, biological, cellular, and molecular properties are all addressed. A special emphasis is given to curcumin’s neuroprotective activities via α7-nAChR mediated mechanism, safety profile, current and upcoming clinical trials for clinical application].
Point 3: Please do a thorough check of the abbreviations, some are repeated or missing (e.g. line 31 then line 34; lines 49, 163, 166, 311, 366).
I agree with this point, I've done a thorough revision of all the abbreviations and missing abbreviations (e.g. DA) that have been added to the text.
Point 4: As the single author of the manuscript please rewrite statements like "In this review, we are discussing and elaborating", "Our recent study on the effects of the natural polyphenol compound provides evidence", "Our in vitro, in silico, and in vivo findings suggest that".
Thank you for your comment; I've completed the necessary work, except in parts of the discussion where there is collaboration work.
Round 2
Reviewer 1 Report
All comments and recommendations have been addressed and the manuscript has been significantly improved.